# What One Gets Is Not Always What One Wants—Young Adults’ Perception of Sexuality Education in Poland

**DOI:** 10.3390/ijerph19031366

**Published:** 2022-01-26

**Authors:** Zbigniew Izdebski, Joanna Dec-Pietrowska, Alicja Kozakiewicz, Joanna Mazur

**Affiliations:** 1Department of Biomedical Aspects of Development and Sexology, Faculty of Education, Warsaw University, 00-001 Warsaw, Poland; 2Department of Humanization of Health Care and Sexology, Collegium Medicum, University of Zielona Góra, 65-046 Zielona Góra, Poland; a.kozakiewicz@cm.uz.zgora.pl (A.K.); j.mazur@cm.uz.zgora.pl (J.M.)

**Keywords:** adolescents, sexuality, sexuality education, sexual health, parents, gender analysis, Poland

## Abstract

There is a discrepancy between the educational needs and the opportunities to obtain reliable knowledge about sexuality in adolescence. This study aimed to assess the conjunctive influence of family and school in shaping this knowledge. Methods: Data were collected retrospectively within a cross-sectional survey conducted in Poland in 2017 (18–26 yrs; N = 595). The respondents’ experiences in terms of the presence and quality of sexuality education (SE) at school and in conversations with parents about related issues were considered. Results: Of all the respondents, 31.1% had no sexuality education classes in school or rated them as useless, 41.5% never discussed sexuality-related topics with parents and both were true for 17.6%. Puberty and contraception were most frequently discussed with parents, while sexual pleasure and masturbation were discussed least frequently. The diversity of topics taken up with parents and the possibility of discussions with mothers turned out to be the most important factors for shaping adolescents’ knowledge. A relationship with the quality of sexuality education at school was additionally revealed for girls, while a relationship with talking to fathers was revealed for boys. Conclusions: Schools and families should work together to strengthen proper sexual development by meeting adolescents’ needs in the field of their knowledge on human sexuality.

## 1. Introduction

The right to a comprehensive sexuality education is one of the sexual rights written into the WHO Universal Declaration of Sexual Rights in 2002 [1,2]. Similar rights are found in the IPPF Declaration of Sexual Rights [3]. Sexual rights are a component of and closely related to fundamental human rights (established by the UN in 1948 as part of the Universal Declaration of Human Rights) [4,5]. The right of children and adolescents to a comprehensive, objective, science-based and culturally sensitive sexuality education is recognized, and its scope is established by international human rights institutions [6]. Therefore, it is one of the unalienable rights of every individual.

In regards to the definition of sexuality education, the broadest and most common is that developed by WHO, UNDP, the Federal Office for Health Education (BZgA) and UNESCO, where comprehensive sexuality education (CSE) is defined as a curriculum-based process of education and learning about the cognitive, emotional, physical and social aspects of human sexuality. In this approach, it should be tackled holistically, treating sexuality education as a subject that enables young people to understand their own sexuality and relationships, and is capable of improving their quality of life [7]. According to UNESCO [8] (p. 69), sexuality education is “an age-appropriate, culturally relevant approach to teaching about sex and relationships by providing scientifically accurate, realistic, non-judgmental information”. Thus, it aims to equip children and young people with the knowledge, skills, attitudes and values that will empower them and enable them to become aware of their health, wellbeing and dignity; develop respectful social and sexual relationships; reflect on how their choices affect their own and others’ wellbeing; and understand and ensure that their rights are protected throughout their lives [2,8,9,10]. Additionally, the International Planned Parenthood Federation (IPPF) defines comprehensive sexuality education specifically as equipping young people with the knowledge, values, attitudes and skills that are necessary and crucial for them to define their own sexuality and enjoy all aspects of it (physical and psychological), individually and in relationships [2,11,12].

Focusing on the main and necessary issues that should be addressed in sexuality education, this knowledge should include topics on the components of human sexuality, feelings and emotions, pleasure, desire, intimacy, gender identification and identity, sexual orientation, anatomy and physiology, healthy and correct sexual development, body image, intimate relationships, sexual activity, abstinence, contraception, procreation, sexually transmitted infections, issues of sexual violence and sexual and reproductive rights [2,10,13,14,15,16,17,18,19].

This knowledge should be provided by different people and institutions. The primary and most natural educational agenda (within informal education) is the family environment [20]. Hence, a prelude to the sexuality education of children are the opinions, attitudes and behaviours of parents in the areas of sexuality and relations with other people [21,22]. The parents and the close family environment are an important source of education, especially for younger children. However, education at home is not always capable of meeting the educational needs of children and adolescents, and thus does not always have a positive impact on their psychosexual development. This problem is exacerbated by the fact that many parents do not address the subject of sexuality, or are reluctant to do so. There may be many reasons for such a situation: lack of sufficient knowledge or vocabulary, shame and fear of “difficult” and deeper questions, concern that these conversations may constitute consent to the child’s sexual activity, their own negative beliefs and attitudes towards sexuality, stereotypical thinking that the child still has time for this type of conversation; tabooing of this topic; and/or their own negative experiences from childhood when they themselves were seeking this kind of knowledge. At a later stage in life, other institutions (kindergarten, school, etc.) start to play an important role and then formal education starts, which only plays a supporting role to informal education [23,24,25,26,27,28,29]. Moreover, young people often prefer to have additional sources of information, other than their parents. This is due to their close emotional relationship, which can make it difficult to engage in sexuality education discussions and cause embarrassment. Peers, the media (especially the internet), culture and the wider society (e.g., medical professionals) play a significant role at this time [28,29,30].

With regard to the situation in Poland, sexuality education continues to be a subject that generates many emotions and controversies. Two strongly polarised and mutually exclusive discourses (conservative and liberal) are present in the public sphere. The dispute in this context mainly concerns the need, purpose, assumptions, influence, forms, content or persons providing the sexuality education. These dilemmas concern not only the name of the subject itself or the age of pupils starting formal education in this field, but also the content and syllabus of such education; the choice and selection of appropriate educational methods, textbooks and educational materials; the preparation of teachers; the status of the subject; and the consent of parents to the participation of pupils in such classes.

The situation is made worse by the fact that in Poland, formalised and institutionalised sexuality education (as well as all other issues concerning sexual and reproductive rights) is mainly a tool of political and ideological struggle. This is evidenced by the social proposal of the bill entitled “Stop Pedophilia”, prepared by the pro-life organization and submitted to the Polish Parliament in 2019, that is still waiting for the voting. It has an implicit manipulation of reality, due to the use of the phrase “sexualisation of children”, which it equates with sexuality education. Consequently, this leads to the intimidation of sexuality educators willing to conduct classes in accordance with the WHO standards, and the proposal to prohibit sexuality education by persons representing NGOs working for sexual and reproductive rights. In addition, the following are clearly visible in the curricula: a lack of neutrality in terms of general worldview, the promotion of family values and a heteronormative model of relationships, the reproduction of stereotypes and atheoreticism (i.e., no reference to current, reliable and evidence-based scientific knowledge; inadequate educational objectives; a lack of substantive criteria for the approval of textbooks; and a lack of reliable research on the effectiveness of education) [31,32]. In practice, this most often means that sexuality education based on the “abstinence only” principle is promoted and preferred. Although there are regulations from the Ministry of Education concerning the aims, contents and effects of sexuality education in Poland, they are mainly based on promoting Catholic values (Dz. U. 1993 poz. 78, z późn. zm.; Dz. U. 2017 poz. 356; Dz. U. 2021 poz. 1533; Dz. U. 2021 poz. 1537). Besides, since the 1970s many initiatives related to the promotion of comprehensive sexuality education have been undertaken by the Polish Family Planning Association, the Federation for Women and Family Planning and the Polish Society of Sexology; however, in recent years, they have been met with increasing resistance from a part of the decision-makers and school principals themselves. Textbooks for teaching this subject also deserve criticism. Most of those included in the ministerial list as approved and applicable should not be used, mainly because the content they present and promote does not serve sexual and reproductive health, as described in the WHO definitions; instead, they often provide information that is contrary to the latest scientific knowledge, repeat myths and stereotypes about human sexuality and issues related to it, present an already outdated (and conservative) family model as the only appropriate one and promote prejudice and discriminatory behaviour towards representatives of different sexes or sexual orientations [33]. Another difficulty in the effective provision of knowledge on human sexuality is the issue of the adequate preparation of teachers, who for the most part merely complete additional training courses, during which the knowledge and content provided to them are not in accordance with the latest scientific research and international standards [34]. It is worth noting that in some Polish schools, there are good and well-prepared teachers, and their pupils are satisfied with the sexuality education given to them. However, this is a minority of cases. Another issue is that at the University of Warsaw in the Faculty of Pedagogy for over 20 years, postgraduate studies in sexuality education have been conducted in accordance with the WHO standards; however, there has been considerable resistance in commissioning those who conduct classes related to them.

The purpose of the study presented below was to provide a retrospective evaluation of the scope and usefulness of sexuality education during adolescence, as expressed by young Poles.

The main research question was: what is the importance of the family for young people in the process of sexual education, and does the family environment determine the knowledge of adolescents about human sexuality? In addition, we decided to answer an additional question: what are the views of young people on sex education at school?

An attempt was made to assess the extent to which discussions with parents concerning human sexuality and positively evaluated school classes had a synergistic effect on the level of knowledge declared during adolescence. Gender differences were also highlighted, including the impact of conversations with the father and mother on daughters’ and sons’ knowledge.

## 2. Materials and Methods

### 2.1. Sample

The survey was conducted in 2017 among a national sample of 2500 adult residents in Poland [35,36]. This was a nationally representative sample from all administrative regions of the country. For the purpose of this paper, the analyses were limited to 595 persons aged 18–26. This group in Poland is entitled to a number of social privileges (e.g., family allowances for students, tax benefits for the employed, etc.). The characteristics of the respondents can be found in Table 1. The analysed group was balanced in terms of gender. The mean age was 20.78 (SD = 2.39) years, and those under 20 were the most represented. Therefore, the majority were students. The percentage of urban and rural residents matched the national statistics [37]. Those living in relationships accounted for 40.5% of respondents, of which 8.6% were in a formal relationship and 5.7% had children. About two-thirds of the respondents had undergone sexual initiation, and the mean age of first intercourse was 17.4 ± 1.67 years. These two rates of sexual initiation were at similar levels among young men and women (*p* = 0.891 and *p* = 0.555, respectively). There were no gender-related differences for the other features that were analysed. Only the percentage of singles was significantly higher in the male group (*p* = 0.045). The sample characteristics showed no missing data, except for 21 cases of undefined employment status (studying, 3.5%). In addition to the above characteristics, the assessment of family situation during adolescence is presented later.

### 2.2. Procedure and Measures

The data were collected in the respondents’ homes in the presence of an interviewer experienced in conducting social quantitative surveys. The research tool consisted of two questionnaires completed within the same meeting using the PAPI (pen and paper personal interview) method. The first questionnaire was filled out during the face-to-face interview, while the second questionnaire was completed by the respondent in person under conditions guaranteeing greater confidentiality. In this part of the survey, the interviewer did not participate actively, but was available to provide clarifications. Both questionnaires were labelled with the same ID number, which made it possible to link the data. The confidential questionnaire was returned in a sealed envelope and placed at random among other envelopes in the urn. Women were interviewed by women and men by men. The interview questionnaire contained 45 questions or blocks of questions, and the confidential questionnaire contained 137 items, totalling approximately 500 variables. Thirty-six variables from both tools were used in this paper. In addition to demographic data (gender, age, place of residence, employment status and relationship status), five questions were selected from the interview questionnaire regarding relationships with the father and mother and living with parents during adolescence, and participation at any stage of education in classes addressing human sexuality and evaluation of the knowledge gained in these classes from the present perspective. Five questions were selected from the confidential survey, constituting a section on sexuality education. They related to opinions about the rationale for sexuality education in school, the postulated character of school sexuality education, having conversations with father and mother about sexuality during adolescence, the topics present in these conversations (13 topics) and evaluations of the influence of ten different people and institutions on the knowledge possessed at that time. The last two sets of questions were optional, to be completed only by those who previously declared having had conversations with their mother, father, or both parents. Consequently, the last question cannot be viewed as a complete profile of sources of knowledge, but only as an assessment of the influence of additional non-family factors. The exact wording of the questions and response categories for school and home-based sexuality education are provided in the following tables.

### 2.3. Analytical Approach

As a first step, secondary variables were defined by combining information from the questions mentioned above. When assessing the usefulness of the knowledge obtained in school, a code of “0”, indicating no such activities, was added to the standard codes (from 1: not at all useful to 5: very useful knowledge). A measure of the diversity of topics covered in conversations with parents was the number of conversations. Here again, no conversations were coded as zero topics. One item relating to conversations with parents was split into two (mother and father).

As a second step, the responses to individual questions were compared for young men and women and the relationship between selected factors was examined separately for both genders using the chi-square test (categorized variables) or the Mann–Whitney test (continuous or ordinal variables). The results of the chi-square test (chi-sq.) are presented along with the degrees of freedom (d.f.) and the significance level “*p*”. The nature of the dependency between the rows and columns of the contingency tables was analysed using adjusted standardized residuals.

As a third step, linear regression models were estimated, in which the dependent variable was the assessment of family influence on adolescents’ knowledge about human sexuality. The explanatory variables were gender, age in years (18–26 yrs.), working or studying (1: yes), being in a relationship (1: yes), opinion about the usefulness of sexuality education in school (range 0–5), the quality of relationship with father and mother during adolescence (range 0–5), conversations with father and mother about sexuality (separately, 1: yes) and the diversity of topics of these conversations (range 0–13). The final models were re-estimated for variables with a proven relationship so as not to reduce the sample size due to missing data. The goodness-of-fit was concluded on the basis of the R-squared coefficient. Significant interactions between the analysed factors were also sought using a general linear model (GLM). The results from this model are presented graphically as marginal means.

### 2.4. Ethical Statement

The methodological framework and logistics of the survey were positively evaluated by the Research Ethics Committee at the Faculty of Pedagogy, University of Warsaw (opinion no. 5/2020). The opinion concerned the thematic scope of the questionnaire and the research procedure, including obtaining the consent of respondents, ensuring their anonymity and giving them the possibility to withdraw their participation in the survey.

## 3. Results

### 3.1. Family Environment in Adolescence

With regard to biological parents or their partners, 94.8% of respondents lived with them during adolescence, of whom 19.0% lived with only one parent. The relationship with the mother (or father’s partner) at that time was rated as very good or good by 80.5% of young people. With respect to fathers (or mother’s partners), the percentage of positive relationship ratings decreased to 65.5%.

Positive relations with the mother during adolescence were declared by 88.1% of men and 81.7% of young women. In the case of the positive evaluation of the relationship with the father, the percentages were 79.5% and 69.4%, respectively. Gender differentiated the opinions about the relationship with the father to a greater extent (chi-sq = 7.5; d.f. = 2; *p* = 0.024) than with the mother (chi-sq. = 5.4; d.f. = 2; *p* = 0.067).

### 3.2. Sexuality Education in School According to Young Adults

The young adults briefly described their own experiences with sexuality education at school and expressed their current opinions about the rationale and postulated character of such education. Almost three-quarters of the respondents participated in classes on human sexuality at school at some point in time. Among the rest, 17.8% declared that they had not attended such classes or could not recall them (7.4%). The percentage of missing data was small, and only 6 out of 595 respondents in this age group refused to answer. More than half (53.6%) of those who participated in sexuality education classes felt that the knowledge gained was useful from their current perspective. However, in this group, 18.7% negatively rated the usefulness of such classes, which is presented in Table 2.

Of the young people surveyed, 75.1% believed that sexuality education classes should be held in schools. Their own experiences projected this opinion (chi-sq. = 29.2; d.f. = 2; *p* < 0.001). The percentage supporting sexuality education in school was 81.1% in the group who had such classes themselves, 69.5% in the group who did not have classes and 50.8% in the group who could not or would not describe their experiences.

Those who supported sexuality education in school also specified what the content should be. Figure 1 shows the distribution of the responses across the sample, taking the lack of support for sexuality education as a separate category. More than one third of the respondents felt that school sexuality education should present different attitudes towards sexual activity, love and relationships, as well as provide sound knowledge on contraception and risky sexual behaviour. A smaller percentage would limit the scope of this education to the last two issues. The most conservative view was represented by 5.5% of respondents, who thought that school sexuality education should convey mostly a religious point of view, knowledge about love and relationships and encourage sexual abstinence before marriage.

In the population of young Poles, gender did not differentiate the above opinions on sexuality education at school (Table 1). The lack of differences also concerned the content that should be conveyed (chi-sq. = 2.7; d.f. = 4; *p* = 0.605). For example, 33.8% of males and 38.5% of young females were in favour of comprehensive education.

### 3.3. Sexuality Education in the Family as Perceived by Young Adults

#### 3.3.1. Taking Up Conversation

Young people answered the question whether during adolescence their parents (guardians) talked to them about human sexuality. No such conversations were indicated by 41.5% of the respondents, 24.7% had conversations with both parents, 27.9% with the mother only and 5.9% with the father only. There were large gender differences in the distribution of responses to this question, confirmed by the analyses of the standardized residuals (all > 1.96). Boys were more likely than girls to have conversations with their father and with both parents, and were also more likely not to have conversations with their parents. In the case of girls, conversations with the mother only predominated with the highest positive residual (Table 3).

It was shown that the frequency of conversations about sexuality was related to the quality of the relationship with parents during adolescence (Table 4). As the relationship improved, the rate of not having such conversations declined significantly. The relationship was strongest with respect to girls’ relationships with their mothers and weakest with respect to their relationships with their fathers. It should be noted that in the case of poor relationships with parents, the percentage of boys not engaging in conversations with their parents increased dramatically (to 71.4% for poor relationship with the father).

Summarizing the results of school and home-based sexuality education, it is worth noting that among the young adults surveyed in Poland in 2017:−31.1% had no sexuality education classes at school, or had classes rated as not useful;−41.5% did not talk to at least one parent about sexuality during adolescence;−17.6% had two of the above risk factors for lack of access to sexuality education.

#### 3.3.2. Topics of Conversation with Parents

Those who reported having conversations with their parents during adolescence about topics related to human sexuality indicated which topics were discussed. The ranking of topics is shown in Table 5 for the total group and by gender of the respondent. Individual topics were addressed with a frequency ranging from 14.3% to 50.6% when considering percentages calculated over the whole group. The topics of satisfaction with sexual life and masturbation were raised least frequently, and the topics of sexual maturity, pregnancy prevention and adolescent love were raised most frequently. Differences related to the gender of the respondent were shown for all 13 topics. Only the topic of masturbation was more frequently discussed with boys than with girls. The largest gender differences were noted for puberty and masturbation. The gender differences were influenced by the fact that boys were less likely to talk to their parents about sexuality. If those who did not talk to their parents about sexuality are ignored, the frequency of individual topics in conversations with boys and girls is similar. Gender-related differences persisted with respect to sexual maturity (*p* = 0.032), masturbation (*p* < 0.001) and risky online behaviour (*p* = 0.033).

A measure of the diversity of conversations undertaken with parents can be seen in the number of topics addressed. Looking at the median, it was shown that half of the persons talked about seven or more of the 13 topics given. No differences related to respondent gender were shown here (*p* = 0.717; Mann–Whitney nonparametric test).

### 3.4. Determinants of Family Impact on Adolescents’ Knowledge about Human Sexuality

#### 3.4.1. The Impact of Family on Adolescents’ Knowledge about Human Sexuality

The 18–26 year-olds also rated the impact of various factors (i.e., people, institutions) on their knowledge of human sexuality experienced during adolescence. The question was asked conditionally if their parents talked to them about these topics at all (N = 323). Only data about the family impact are presented below. The family influence index was estimated for the whole sample of 595 individuals, taking the value of zero for those who did not talk to their parents, resulting in a mean score of 2.32 (SD = 1.82) and significant gender-related differences (*p* < 0.001; nonparametric test). This was largely influenced by the higher percentage of girls talking to their parents.

In the group of adolescents who talked to their parents about sexuality, gender-related differences are less evident (Table 6), with higher a mean score (3.38 ± 1.04).

The mean impact score of each person or institution is shown in the Appendix A (Table A1), with higher scores indicating the greater influence of a particular factor. Peers and the internet were rated highest as additional sources of knowledge, and church and medical staff were rated lowest. The gender differences were mostly statistically insignificant. Young women only rated the influence of other media (apart from the internet) higher, and young men rated the influence of school pedagogical staff (apart from teachers directly teaching about sexuality) higher.

#### 3.4.2. Determinants of Family Impact in Girls and Boys

An attempt was made to assess the factors that influenced the variability in the assessment of family influence on the knowledge about human sexuality possessed during adolescence. Table 7 shows the results of the linear regression models, estimated separately for girls and boys. All the factors described above were considered. However, in the final models, only those that were significant in at least one gender were included.

Each model explained 58% of the variability in the assessment of family influence on adolescents’ knowledge. All the regression parameters were positive; that is, the presence of a factor or an increase in a factor resulted in improved knowledge. The most significant predictor appeared to be the diversity of the topics covered in conversations with parents, followed by having conversations with the mother. The association with the former factor was stronger in the boys’ group and with the latter in the girls’ group. Having conversations with the father was significant only in boys, while age and quality of school sexuality education influenced the variability in the girls’ knowledge scores.

In addition, the interactions between the analysed factors were tested using the general linear model (GLM). School sexuality education scores were recoded into three levels (1—no education; 2—score 1 to 3; 3—score 4 to 5).

Figure 2 illustrates the accumulation of protective factors (high quality of school sexuality education and talking to the mother), which were more evident in girls. The three-way interaction was significant at the *p* = 0.005 level. In case of boys who talked to their mother about sexuality, their knowledge about sexuality was stable regardless of the quality of school sexuality education. However, this knowledge improved with higher levels of school sexuality education in the group of boys who did not talk to their mother about these topics. These analyses confirm that the gender factor affects adolescents’ knowledge in interaction with other factors.

## 4. Discussion

The results of many studies have indicated that adolescents show considerable interest in comprehensive sexual education, and access to sexual education is commonly perceived as a basic right [16]. Moreover, in countries liberally oriented towards sexual education, the later initiation of adolescents has been reported [38]. Referring to the topic of the study, it can be stated that not all young adults surveyed in Poland have had the opportunity to obtain adequate knowledge at home or at school during adolescence. It was shown that 17.6% did not have SE classes at school, or they were not useful, and at the same time there were no discussions with parents on these issues. The findings also indicated that there was little diversity in the topics discussed with parents. The work presented here correlates with an earlier report on a population of 18–49 year-olds [35]. In many ways, the opinions of people aged 18–26 turned out to be more positive than those of older people. It is possible to speak of a very slow generational change. People aged 27–49 declared access to knowledge on sexuality during adolescence even less frequently than younger people and discussed these issues with their parents even more rarely.

There is general agreement that parents are the first and most important teachers for their children. It has been pointed that a “quality parent” is one who strives to facilitate the child’s learning in a variety of ways and at different times. These parents see their children as apprentices in life who need to be guided by adults, and adults transform and socialize their children so that they can reach their full potential [39].

Research on sexual communication points out the role of gender, psychological factors and family dynamics in the effectiveness of sexuality education. It has been found that most of the communication on sexual issues comes from the mother. In contrast, boys believe that the communicated content is mainly directed at girls’ experiences. Therefore, boys use other sources (i.e., peers, media and the internet) to educate themselves on sexuality issues. Although parents want to talk to their children about issues related to sexual behaviour, they feel embarrassed and uncomfortable, and have neither the skills nor the knowledge to do so [40]. In our study, a lack of sexuality education talks was indicated by 41.5% of the respondents; 24.7% talked to both parents, 27.9% talked to the mother only and 5.9% talked to the father only. Peers and the internet were judged best as additional sources of knowledge, while the church and medical personnel were judged as the worst. Young women only rated the additional influence of other media (apart from the internet) higher, while young men gave a higher rating to the influence of school personnel (apart from teachers actually teaching about sexuality). In addition, there are studies available that report men rating pornography as a more important source of sexual information than women. This gender difference may reflect the fact that men are more likely to view sexually explicit material and do so more frequently than women [30]. It is also worth noting that in the Australian study, among the most frequently observed sources of support for sexuality education, 60% of students chose “parent/guardian”, then “friend” (46%), then “health service” and “internet” (both 39%) and finally “teacher” (32%) [27]. In another study among Chinese students, 72.5% of respondents said they would prefer to receive sexuality education from internet sources. In the post-COVID-19 era, online learning has become the standard, making it a particularly preferred way to teach thorny and potentially embarrassing topics [41]. The results presented here highlight the importance of sexuality education provided by parents, as well as the importance of other sources.

Sexual communication in the family presents unquestionable challenges. Research indicates that girls are more likely to turn to parents and boys to peers about sexuality [42]. In one study, university students and their parents were asked if they ever had a meaningful discussion about sex in the family environment. More than half of the students answered in the negative; however, in 60% of those cases, one or both parents said they had conducted significant discussions. In families where there were parent–child disagreements, the biggest differences were on the topics of sexual cohabitation, reproduction, birth control, sexually transmitted diseases, homosexuality and sexual abuse. This was true for both mothers and fathers. Mothers were more likely than fathers to have discussions that daughters felt were important, while fathers were more likely to have these discussions with their sons. The parents who indicated that they had important conversations about sex with their parents while growing up were significantly more likely to have discussions that their own children found meaningful. Nevertheless, the results show that many parents highly underestimate the extent of the actual information about sexuality that their children want learn [43]. These findings support the results of studies measuring parental involvement in sexuality education, which suggest that adolescents want more interaction with their parents about sexuality, but that the approaches identified by parents conflict with ideas generated by adolescents [44]. It is also worth noting that, compared to fathers, mothers rated sexual health communication with their adolescents as more comfortable and had a greater sense of self-efficacy in this area. Mothers also discussed sexual health with their children more frequently and more broadly than did fathers [45]. Our own research shows that as relationships improved, the rate of not discussing sexuality decreased noticeably. In the case of bad relationships with parents, the percentage of boys not engaging in conversations with their parents about sexual topics increased sharply (to 71.4% for bad relationships with their father). Moreover, adolescents who discussed more sexual health issues with their parents and best friends were also more likely to talk about sex in in their early dating relationships [46]. It is emphasized that theory-based, developmentally appropriate and comprehensive sexuality education programs that include parental involvement may be effective in delaying vaginal sex among secondary school students. Parental involvement is especially important for boys because it can lead to earlier and more frequent conversations with sons [47].

Therefore, in the process of sexuality education, it is extremely important to emphasize the relational factor, which plays a significant role in the transmission of sexual content. Moreover, in the long-term perspective, it allows young people to create more satisfying interpersonal relationships. It is also impossible to minimize the role of proper communication aimed at the needs of children and adolescents.

In a sexuality education that is democratic, community-based and collective, everyone would receive medically accurate information about contraceptives and their availability, and pleasure would be discussed in the context of pleasing the other person and making them feel comfortable. Students would research topics related to sex in society to become responsible citizens. They would receive knowledge of societal concerns about pornography, sexual violence, consent, sex work, objectification, relating and sexualization of children [48]. However, available research in this area shows that these topics are not sufficiently addressed.

In one study, high school students completed questionnaires describing the frequency and importance of communication with their mother and father about 20 different sex-related topics. The study identified four major domains of sex-related topics: social development and concerns, sexual safety, experiencing sex and masturbation. The adolescents reported infrequent communication that varied by domain and the gender of the parent and adolescent. When communication did occur, the first two domains were most often involved. Mothers have been reported to communicate more frequently about sexuality than fathers, and girls have been reported to communicate more frequently than boys [49]. In another study, while assessing the parents’ perceptions (n = 374) of the characteristics, content and comfort of discussing sexual concerns with their adolescents, almost all parents (94%) admitted that they had talked to their adolescents about sex. Two-thirds (65%) said they were comfortable talking with their teenage children about sexual issues. From a list of 17 potential topic areas for sexual communication, the parents were most likely to talk to their teenagers about the responsibilities of being a parent (46%), sexually transmitted diseases (40%), dating behaviour (37%) and not engaging in sexual activity until marriage (36%). Almost all parents (92%) believed that sexuality education should include information about birth control, including condoms [50]. A review that was designed to present what is known about the prevalence and effectiveness of sexuality education programs, thereby informing better public policy making in this area, is also worth noting. Twenty of the presented reviews focused primarily on reducing risky behaviours (e.g., sexually transmitted diseases and unwanted pregnancies), while at the same time neglecting topics such as desire and pleasure [51].

Our own research shows that puberty, pregnancy prevention and adolescent love were most frequently discussed with parents. The least frequently discussed topics were satisfaction in sexual life and masturbation; however, the topic of masturbation was more frequently discussed with boys than with girls. It is also worth noting that in the ranking of topics discussed with parents, pornography and sexual violence occupied a distant position, which indicates that these topics are not popular. Currently, specialists point to the need to treat sexual violence not only as a crime, but also as a public health problem [52]. Appropriate education should not be limited to protection from victimization. Primary prevention targeting potential perpetrators of violence is considered a recognized course of action. Comprehensive sexuality education should pay attention to documented factors that increase the risk of being a perpetrator of violence, including: (1) gender and violence-related risk factors; (2) child abuse-related risk factors; (3) sexual behaviour-related risk factors; and (4) social-emotional intelligence-related risk factors [53].

In our study, half of the participants talked about seven or more of the 13 topics listed. Research indicates that adolescents have a significant interest in understanding the feelings and perceptions of the opposite sex and acquiring this knowledge can be beneficial in building emotionally and physically satisfying relationships [54]. In order to increase the sexual and relational competence of adolescents, sexuality education programs, as well as content provided directly by parents, should also include aspects of sexuality related to pleasure, desire and sexual violence, as well as masturbation for girls, which is an aspect that—perhaps due to conditions related to the lower social acceptance of the autoerotic behaviour of girls—is still discussed less often. In light of the results of our own research, the least discussed topics in Polish families were sexual pleasure (according to men) and masturbation (according to young women), as well as sexual violence.

Based on the premise that sexual experiences and sexual communication are shaped by the messages we receive in our youth, the impacts of sexuality education received at home and in school are complementary. The data showed that more than half (53.6%) of those who participated in sexuality education classes felt that the knowledge received was useful from their current perspective. In this group, 18.7% negatively evaluated the usefulness of the discussed classes. There are other models of sexuality education in scientific literature that could increase the usefulness of these classes. One of these models of comprehensive sexuality education goes beyond education about reproduction, risks and diseases, and includes respecting the sexual rights of boys and girls. Complex and holistic sexuality education is guided by a comprehensive curriculum that focuses on human rights and gender equality, while influencing efforts to prevent HIV, sexually transmitted diseases, unintended pregnancies and gender-based violence [55]. It is also worth noting that a short-sighted focus on abstinence all the way to the exclusion of problematizing traditional gender roles and sexual scripts will not facilitate the transformation toward greater sexual health and empowerment. Moreover, a longitudinal study among adolescents found that greater sexual knowledge predicted fewer rape-supportive beliefs six months later [56]. Furthermore, the limited available research has shown that students value topics that reflect positive sexuality, including healthy relationships and sexual pleasure [57]. This fact is supported by the results of our own study, in which more than one-third of the respondents felt that school sexuality education should present different attitudes toward engaging in sexual activity, love and relations in a relationship, and should also provide sound knowledge about contraception and risky sexual behaviour. This research supports the importance of prioritizing comprehensive sexual health education that covers a wide range of topics that include educational content about accessing local services and resources as well as acknowledging and understanding both the positive and negative aspects of sexual activity and relationships. The results also highlight the importance of different sources of access to information for youth, social influences and safe learning environments in the context of sex education [58].

Combining the dimensions of sexuality education provided by parents and school, it is worth noting that despite the overwhelming support for the inclusion of sexual health education in schools, few parents perceived the sexual health education that their child/children received as very good. Parents supported teaching about important new areas, such as sexuality, sexting and media skills related to sexual content on television, pornography and in advertising [59].

The limitations of our analysis result from the fact that we used an element of a larger study aimed at the adult population and various issues related to its health and sexuality. The number of available questions on school and home-based sexuality education was limited and the extracted sample of 18–26 year-olds was not very large. However, being limited to an age-homogeneous group of young adults is an advantage of this study. These are individuals who grew up in similar circumstances, have parents of similar age and have a good memory of adolescence (which offsets recall bias). The design of the questionnaire included a number of filters to avoid questions, which streamlined the survey process but resulted in further limitations. In view of the filters used, for example, it was not possible to fully assess the scope of non-family sources of knowledge about sexuality for adolescents who did not discuss the indicated topics with their parents, including the scope of school education. Moreover, family communication was analysed with a focus on conversations about human sexuality, their frequency, involved family members (mother, father or both) and the topics discussed. Other studies also paid attention to the conformity aspect, defined as the degree to which children feel obligated to hold the same beliefs as their parents. This approach, grounded firmly in family communication patterns theory, should be used in future research on sexual education and its impact on the behaviours and decisions undertaken by youth [60].

However, this study has a number of advantages that compensate for the above weaknesses. The data are still up-to-date, and in the light of the knowledge available to us, this is one of the few quantitative studies combining the topics of school and home sexual education in Polish conditions, with a strong emphasis on the characteristics of the latter.

## 5. Conclusions

The results of our research indicate the limited access of Polish adolescents to sexuality education. It was found that girls receive more information on sexuality than boys and the thematic range of the discussed content, both within the school and the family, is limited. There is a real need for supplementary, broad-perspective, comprehensive and science-based sexuality education.

In view of the unfavourable atmosphere surrounding school sexuality education in Poland, strengthening the role of parents seems to be an important course of action. However, the results of the analyses indicate that the family together with the school should take care of the proper sexual development of adolescents by responding to their demand for knowledge. Our findings also support prioritizing the training of teachers and parents to increase their comfort with the subject matter and improve the communication of sexuality education content.

## Figures and Tables

**Figure 1 ijerph-19-01366-f001:**
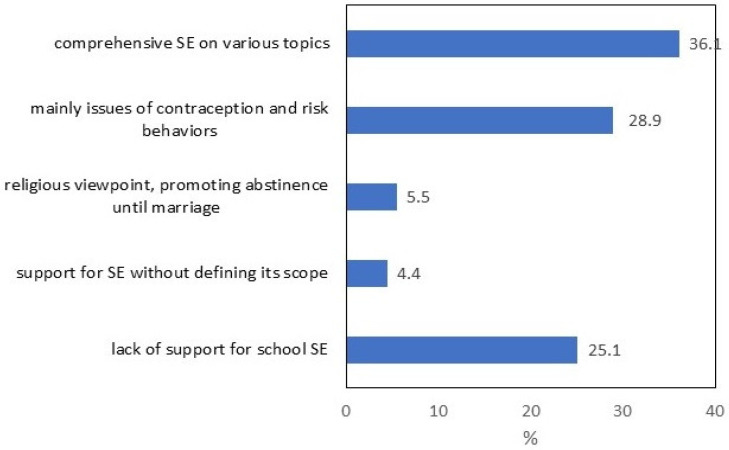
Young people’s preferred scope of school sexuality education (SE).

**Figure 2 ijerph-19-01366-f002:**
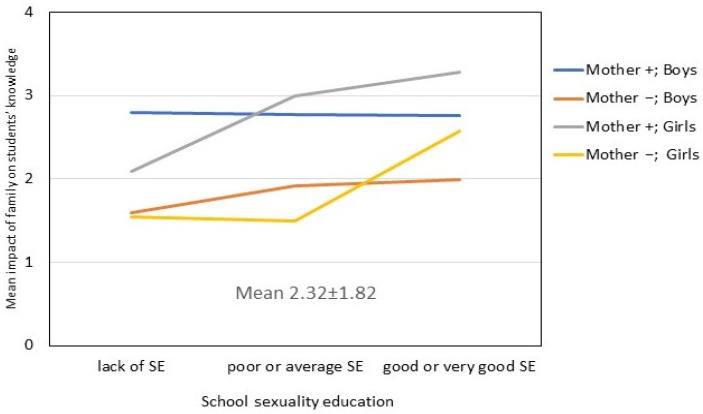
The interaction between gender, quality of school sexuality education (SE) and talking to mother as predictors of adolescents’ (N = 595) knowledge about sexuality (marginal means from GLM model).

**Table 1 ijerph-19-01366-t001:** Sample characteristics (all data presented as percentages).

Variable	Categories	Total N = 595	Male N = 299	Female N = 296
Age	18–20 years	62.5	62.5	62.5
	21–23 years	20.0	21.4	18.6
	24–26 years	17.5	16.1	18.9
Employment	Working	40.4	42.0	38.8
	Studying	47.4	47.2	47.6
	Other *	12.2	10.8	13.6
Place of living	Large cities	25.4	23.7	27.0
	Small towns	33.4	35.8	31.1
	Rural areas	41.2	40.5	41.9
Living with whom	Alone	5.4	5.4	5.4
	With at least one parent	76.1	77.9	74.3
	With other people but not parents	18.5	16.7	20.3
Staying in relationship	Single	59.5	63.5	44.6
	Formal or informal relationship	40.5	36.5	55.4

* Unemployed, on pension, or missing data.

**Table 2 ijerph-19-01366-t002:** Opinions of 18–26 year-olds about sexuality education in school (%).

Question	Responses	Total	Male	Female	*p*
At any stage of your education in school, did you participate in classes that addressed human sexuality?	Yes	72.3	68.2	76.4	Chi-sq. = 5.31 d.f. = 2 *p* = 0.070
	No	17.8	21.1	14.5	
	Cannot remember	7.4	10.7	9.1	
	Missing data	2.5			
From your current perspective, how do you assess the usefulness of the knowledge you gained there?	Completely useless	6.6	15.6	21.4	Chi-sq. = 2.38 d.f. = 2 *p* = 0.305
	Useless	12.1			
	Neither useful nor useless	27.7	28.6	26.8	
	Useful	43.7	55.8	51.8	
	Very useful	9.9			
In your opinion, should sexuality education classes be taught in schools?	No	23.7	23.4	24.0	Chi-sq. = 3.57 d.f. = 2 *p* = 0.168
	Yes	75.1	74.6	75.7	
	Missing data	1.2	2.0	0.3	

**Table 3 ijerph-19-01366-t003:** Talking to parents about sexuality issues (% and [adjusted standardized residuals]).

Parent Talked to	Male	Female	*p*
Mother and father	32.1 [4.2]	17.2 [−4.2]	Chi-sq. = 103.4 d.f. = 3 *p* < 0.001
Only mother	10.0 [−9.8]	45.9 [9.8]	
Only father	9.7 [4.0]	2.0 [−4.0]	
Parents were not talked to	48.2 [3.3]	34.9 [−3.3]	

**Table 4 ijerph-19-01366-t004:** Young adults (%) who have not talked to their parents about sexuality in adolescence by gender and relationship with parents at the time.

Parent and Respondent Gender	Quality of Relationship with Parent			*p*
	Poor or Very Poor	Neither Poor nor Good	Good or Very Good	
Boys—did not talk to mother	60.0	58.6	46.2	0.002
Girls—did not talk to mother	58.3	59.0	26.3	<0.001
Boys—did not talk to father	71.4	61.0	42.8	0.003
Girls—did not talk to father	43.5	35.2	26.3	0.020

**Table 5 ijerph-19-01366-t005:** Topics of conversation with parents.

Topic	Responses (N)	% in Relation to the Total Sample (N = 595)			*p*
		Total	Male	Female	
Sexual satisfaction, orgasm	327	14.3	13.0	15.5	0.006
Masturbation	326	15.5	18.7	12.2	<0.001
Sexual violence	328	23.7	20.7	26.7	0.008
Pornography	328	23.9	18.4	29.4	0.001
Risky behaviour on the internet (sending photos, meeting strangers)	327	25.0	19.1	31.1	0.001
Sexual orientation	331	27.9	26.1	29.7	0.004
Morality/ethics in sexual relationships	329	28.7	23.4	34.1	0.002
Risk sexual behaviour (e.g., under the influence of alcohol and drugs)	333	32.9	29.4	36.5	0.006
Sexually transmitted diseases (including HIV)	332	37.3	33.4	41.2	0.008
Sexual debut	329	39.8	35.5	44.3	0.017
Juvenile attraction	332	44.5	37.1	52.0	0.001
Contraception	333	45.2	38.8	51.7	0.004
Puberty	335	50.6	42.8	58.4	<0.001

**Table 6 ijerph-19-01366-t006:** Family impact on adolescents’ knowledge about sexuality among those who talked to their parents (%; total N = 323).

Level of Impact	Total	Male	Female	*p*
1—very low or none	6.8	5.8	7.6	Chi-sq. = 4.5 d.f. = 4 *p* = 0.345
2	9.9	12.2	8.2	
3	25.7	25.2	26.1	
4	34.7	38.1	32.0	
5—very high impact	22.9	18.7	26.1	
Mean level	3.38 ± 1.04	3.42 ± 0.93	3.35 ± 1.13	

**Table 7 ijerph-19-01366-t007:** Determinants of the level of family impact on sexuality knowledge during adolescence obtained from linear regression models.

Independent Variable (Range)	Male (N = 293)		Female (N = 293)	
	Beta **	*p*	Beta **	*p*
Age in years (18–26)	0.011	0.768	0.109	0.006
Talking to mother about sexuality *	0.195	0.000	0.315	0.000
Talking to father about sexuality *	0.121	0.020	0.015	0.707
Diversity of topics (0–13)	0.530	0.000	0.407	0.000
Quality of school SE (0–5)	0.057	0.155	0.249	0.000
R-sq	0.576		0.580	

* 1—yes; 0—no; ** standardized beta.

## Data Availability

The data are owned by Warsaw University and are not to be made freely publicly available.

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
