# Peer review of "What One Gets Is Not Always What One Wants—Young Adults’ Perception of Sexuality Education in Poland"

_ijerph, 2022, doi:10.3390/ijerph19031366_

Round 1
Reviewer 1 Report
Thank you for the opportunity of reading this interesting article. I have only minor remarks to it.
Introduction
Please pay attention to the order of citations in parentheses.
Line 99 – please add a reference to the bill.
Line 111-112 – please add a reference to the regulation.
Lines 135 if this is the one of … please add references to the other research
What was the main research question?
Material and methods
The survey was conducted in 2017 and the Ethical Committee approval is from 2020. Why is that?
Are the survey and survey database publicly available somewhere?
Results
Rows in tables should be separated somehow. The tables are hard to read.
Counting the standardized residuals would be useful in table 3.
Discussion
I think that the limitation should also include the fact that the questionnaire was completed in the company of an interviewer “The first questionnaire was filled out by the interviewer on the basis of the conducted interview, while the second questionnaire, containing more sensitive questions related to sexual life, was filled out individually. The first questionnaire was filled out by the interviewer on the basis of the conducted interview, while the second questionnaire, containing more sensitive questions related to sexual life, was filled out individually.” It could somehow influence the answers given on such a sensitive topic.
Reviewer 2 Report
Overall, the paper is interesting and I believe it may be of value to readers. Still, there are some points that need to be addressed.
- You need to check that all the documents mentioned are referenced. There are some that do not.
- It would be advisable to provide a separate introduction to the theoretical framework, as it is not clear what the objective of the article is and how this will be achieved. It is usually possible to make these points in the introduction to give readers more clarity.
- It is necessary to explain why the survey is from 2017. You do not have something newer. I am concerned about the use of data from more than 5 years ago, as I find it unreliable... especially on a topic that is so current and changing so rapidly from one generation to the next. Also, most of the population are students... this changes a lot in a few years.
- It would be useful to include the questions of the instrument or the reference of the instrument to be able to access it, in order to get an idea of what is being asked.
- I am somewhat suspicious, and I think it should be considered as a limitation of the research, the fact that the instrument has a self-administered part and another part filled in by someone based on an interview. There is no guarantee that the interview gave accurate information or did not require interpretation, which affects reliability.
- I wish the references could be updated a bit. Only 40% are from the last 5 years, and considering the subject matter, it would be important that the sources are very current.
Round 2
Reviewer 2 Report
Thank you very much for considering most of the comments and implementing them in your text. I value:
a. That you have updated the references
b. That you have made the objective and research questions clearer and more visible.
c. That you have clarified the reason for the survey year.
d. That you have clarified the way in which the instrument was applied.
The paper was already good, but these changes make it much better appreciated.